

# NT-proBNP serves as a prognostic marker for adverse outcomes in severe immune checkpoint inhibitor-associated myocarditis

Cheng He[1,2], Linjuan Xu[2], Zhihong Zhang[2] and Jiong Wang[1]

[1] Department of Geriatric Respiratory and Critical Care Medicine, The First Affiliated Hospital of Anhui Medical University, Hefei, Anhui Province, China
[2] Department of Oncology, The First Affiliated Hospital of USTC, Division of Life Sciences and Medicine, University of Science and Technology of China, Hefei, Anhui Province, China

Corresponding authors
Cheng He, hc198074@outlook.com
Jiong Wang,
wangjiong7286@163.com

## ABSTRACT

**Background**. Immune checkpoint inhibitors (ICIs) have transformed cancer treatment but carry risks of rare, life-threatening immune-related adverse events, particularly myocarditis. Prognostic biomarkers and optimal management strategies for severe ICI-associated myocarditis remain poorly defined.

**Methods**. This single-center retrospective cohort study analyzed 71 patients diagnosed with ICI-associated myocarditis among 7,157 ICI-treated individuals at a tertiary center (August 2018–August 2024). Myocarditis severity was graded per American Society of Clinical Oncology (ASCO) guidelines. Cardiac biomarkers, including NT-proBNP and troponin T, were assessed. Binary logistic regression identified predictors of mortality. Treatment protocols and immunotherapy rechallenge outcomes were evaluated.

**Results**. Severe myocarditis (Grades 3–4) occurred in 33 patients (46.5%), with an overall mortality rate of 54.5% in this subgroup (18/33). NT-proBNP levels were significantly elevated in fatal cases *versus* survivors (median: 13,804 *vs.* 4,050 pg/mL; $P < 0.001$) and independently predicted mortality risk (odds ratio (OR) 4.3, 95% confidence interval (CI) [1.2–21.9]; $P = 0.023$). A multimodal regimen combining plasmapheresis with high-dose corticosteroids, intravenous immunoglobulin, and mycophenolate mofetil was associated with improved survival. Among nine patients rechallenged with immunotherapy, seven (77.8%) tolerated subsequent cycles without recurrent immune toxicity, while two with prior Grade 2 myocarditis experienced symptom recurrence.

**Discussion**. Elevated NT-proBNP emerges as a critical prognostic marker for risk stratification in severe ICI-associated myocarditis. Immunotherapy rechallenge appears feasible in select patients but warrants caution in those with prior moderate-grade myocarditis. These findings advocate for biomarker-guided escalation of therapies and shared decision-making frameworks to balance oncologic efficacy with cardiovascular safety.

## INTRODUCTION

Cancer remains the leading cause of death globally, with most cancer-related deaths attributed to metastasis (*Kiri & Ryba, 2024*). Immunotherapy has revolutionized cancer treatment by significantly prolonging survival in patients with advanced malignancies (*Zhang & Zhang, 2020*). Immune checkpoint inhibitors (ICIs), the most established immunotherapies, are widely used across various cancers. Following the approval of ipilimumab-the first CTLA-4 inhibitor for unresectable or metastatic melanoma-multiple PD-1/PD-L1 inhibitors, including nivolumab and pembrolizumab, have been approved worldwide.

Despite remarkable advances in cancer therapy with ICIs, their broader success is limited by immune-related adverse events (irAEs) (*Conroy & Naidoo, 2022*). These toxicities may affect any organ system in ICI-treated patients and are often unpredictable (*Conroy & Naidoo, 2022*). While most irAEs are mild or self-limiting, severe and potentially fatal complications, though rare, have been documented (*Sullivan & Weber, 2022*). Notably, ICI-associated myocarditis presents unique management challenges and exhibits substantially higher mortality rates compared to other irAEs (*Munir et al., 2024*; *Barzkar, Miles & Mehta, 2024*). The low incidence of this condition (0.04%–1.14%) has resulted in limited studies addressing its clinical course (*Palaskas et al., 2020*; *Nielsen et al., 2024*). We conducted a retrospective analysis of 71 patients diagnosed with ICI-associated myocarditis to characterize its clinical features and treatment outcomes, offering insights to guide future research on mitigating this rare yet critical immunotherapy complication.

## MATERIALS & METHODS

### Patient population

This retrospective cohort study included consecutive patients diagnosed with ICI-associated myocarditis following immune checkpoint inhibitor (ICI) therapy at the First Affiliated Hospital of the University of Science and Technology between August 2018 and August 2024. Data were extracted from electronic medical records and included demographics (age, sex), medical history (smoking status, comorbidities), clinical manifestations (symptoms, signs), laboratory results (cardiac troponin T, NT-proBNP), diagnostic imaging (ECG, echocardiography), ICI regimen details (drug type, dosing schedule), time-to-onset of myocarditis post-ICI initiation, treatment strategies, and clinical outcomes. The diagnosis of ICI-associated myocarditis was established by a multidisciplinary team (MDT) comprising cardiologists, oncologists, and clinical pharmacologists. A multimodal diagnostic strategy was employed, integrating clinical presentation, cardiac biomarkers, and imaging modalities (echocardiography and cardiac MRI [CMR]). Cardiac MRI (CMR) was performed in 34 patients (47.8% of the cohort), with all studies demonstrating features consistent with myocarditis, including late gadolinium enhancement (LGE) and/or T2-weighted hyperintensity. Endomyocardial biopsy (EMB), though considered the diagnostic gold standard for myocarditis, was not performed in any patient due to two primary reasons: (1) in 34 patients (47.8% of the cohort), CMR provided diagnostic features (*e.g.*, LGE, T2-weighted hyperintensity) that rendered invasive confirmation unnecessary in clinical

practice; and (2) in 37 patients, clinical urgency or contraindications (*e.g.*, pacemakers, severe renal impairment) precluded CMR or EMB. For these patients, diagnosis relied on a combination of clinical criteria, biomarker elevation, ECG, and echocardiographic findings after rigorous exclusion of other etiologies. The study protocol was approved by the Clinical Research Ethics Committee of the First Affiliated Hospital of the University of Science and Technology (Protocol Number: 2024-RE-288). The requirement for informed consent was waived owing to the retrospective nature of the study. The follow-up period for the entire cohort began from the date of myocarditis diagnosis and continued until the date of death, loss to follow-up, or the end of the study (August 2024), whichever occurred first. The median follow-up duration was 30 days (interquartile range [IQR]: 30–240 days). For patients who were rechallenged with ICIs, follow-up began from the date of rechallenge and continued until the occurrence of irAEs, recurrence of myocarditis, death, or the end of the study. The median follow-up duration after rechallenge was 17 months (interquartile range [IQR]: 4–35 months).

## The grades of ICI-associated myocarditis

The severity of ICI-associated myocarditis, according to the American Society of Clinical Oncology (ASCO) Clinical Practice Guideline, is defined as Grade 1 when there are elevated cardiac biomarkers without clinical symptoms or ECG abnormalities; Grade 2 when there are elevated cardiac biomarkers accompanied by mild symptoms or new ECG changes, not including conduction delays; Grade 3, elevated cardiac biomarkers with moderate symptoms or new conduction delays; and Grade 4, when there is moderate to severe cardiac decompensation requiring intravenous medications or interventions, including life-threatening conditions (*Schneider et al., 2021*). To effectively categorize the severity of myocarditis, myocarditis was defined as <3 (Grades 1 or 2) as non-severe myocarditis, whereas myocarditis Graded 3 or higher (Grades 3 or 4) was defined as severe myocarditis.

## Statistical analysis

All statistical analyses and graphical representations were performed using GraphPad Prism 9.0 (GraphPad Software, Inc., La Jolla, CA, USA) and R version 4.4.2 (R Foundation for Statistical Computing, Vienna, Austria). Continuous non-normally distributed variables (*e.g.*, biomarker levels) were reported as median with interquartile range (IQR) and compared using Mann–Whitney $U$ tests in GraphPad Prism. Categorical variables (*e.g.*, mortality rates) were expressed as absolute frequencies with percentages (%) and analyzed using chi-squared or Fisher's exact tests.

Binary logistic regression analyses were conducted in R (v4.4.2) using the logistf package with Firth's bias-reduced penalized likelihood method to address potential separation issues in small sample sizes. Results were reported as odds ratios (OR) and 95% confidence intervals (CI). Model assumptions were validated through Hosmer-Lemeshow goodness-of-fit tests ($P > 0.05$) and visual inspection of residual diagnostic plots. Forest plots for multivariable analysis were generated using the forestplot package in R to ensure publication-quality visualization. A two-tailed $\alpha$-level of 0.05 defined statistical significance.

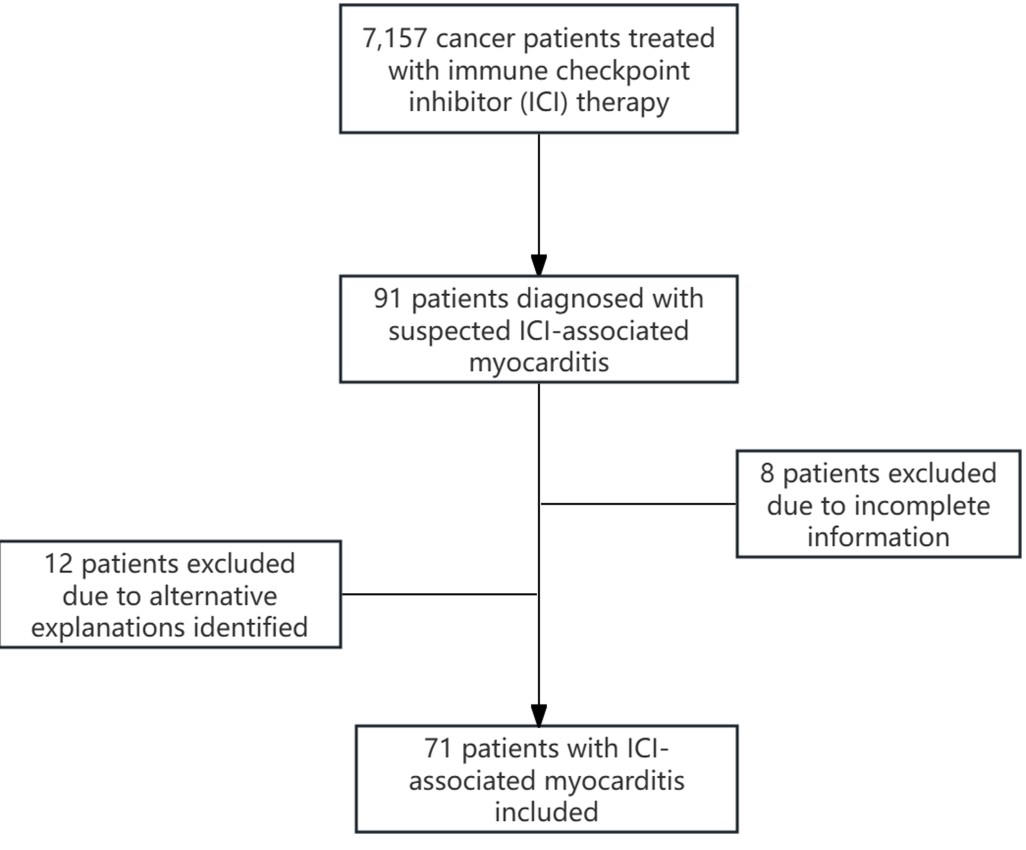

**Figure 1** **Flowchart of patient selection process.**

## RESULTS

### General clinical characteristics of patients with ICI-associated myocarditis

Between August 2018 and August 2024, 7,157 cancer patients received ICI therapy. Among these, 91 individuals were clinically suspected to have ICI-associated myocarditis. Following MDT deliberation, 12 cases were excluded because of alternative explanations for their elevated cardiac biomarkers or chest discomfort. Additionally, eight patients were excluded because of incomplete information. Finally, 71 patients diagnosed with ICI-associated myocarditis were included in this study (Fig. 1). The general clinical characteristics of the patients with ICI-associated myocarditis are summarized in Table S1. Among them, more than half were non-smokers. Most patients were male and over 65 years old.

This study included 15 tumor types, with lung cancer being the most prevalent (40.9%), followed by gastric cancer (14.1%) and esophageal cancer (11.3%). Most patients presented with Stage IV disease and received immunotherapy in combination with other therapeutic agents. In this study, sintilimab and camrelizumab were identified as the most prevalent immune checkpoint inhibitors among the evaluated therapeutic agents.

Upon diagnosis of ICI-associated myocarditis, the majority of patients typically present with clinical manifestations, including chest pain, dyspnea, and palpitations. Notably, one patient exhibited severe abdominal pain as the chief complaint, prompting an emergency department visit. Among all patients diagnosed with myocarditis, thirty patients presented with additional irAEs such as hepatitis, myositis, myasthenia gravis, pneumonitis, and thyroiditis, with myositis being the most common.

ICI-associated myocarditis was primarily observed within the first three months of treatment. However, several cases of ICI-associated myocarditis were diagnosed more than 12 months after the initiation of ICI therapy, with two patients diagnosed nearly two years after the onset of treatment.

## Comparison of clinical characteristics between patients with severe and non-severe ICI-associated myocarditis

To further characterize the clinical course of immune checkpoint inhibitor (ICI) myocarditis, we compared the clinical features between patients with and without severe myocarditis, as detailed in Table 1. No significant differences were observed in terms of sex, smoking status, age, cancer type, or medical history (hypertension and diabetes mellitus) between the two groups. Additionally, there were no significant differences in the types of ICI used or the line of treatment.

However, the incidence of other irAEs, in addition to myocarditis, was higher in the severe myocarditis group (57.6%) than in the non-severe myocarditis group (36.8%). Notably, the proportion of patients with multiple irAEs (two or more) was significantly higher in the severe myocarditis group (24.2%) than in the non-severe myocarditis group (5.3%) ($P = 0.037$).

Myocarditis onset occurred at a median of 50 days (IQR: 29–90 days) after the initiation of immune checkpoint inhibitor therapy (Fig. 2A). Patients with severe myocarditis presented with symptoms at a median of 40 days (IQR: 25–67 days), which was significantly earlier than that in the non-severe myocarditis group, with a median of 60 days (IQR: 30–150 days; $P = 0.028$) (Fig. 2B). Severe myocarditis was associated with markedly elevated cardiac biomarker levels compared with non-severe myocarditis, as illustrated in Fig. S1. All comparisons were statistically significant ($P < 0.001$).

The analysis of cardiac biomarkers' prognostic utility was restricted to patients with severe myocarditis, as no mortality events were observed in non-severe cases ($n = 38$). Within this cohort, patients were stratified into two distinct outcome groups: deceased ($n = 18$) and recovered ($n = 15$). Of the various cardiac biomarkers evaluated, NT-proBNP demonstrated a statistically significant elevation in median levels among patients who succumbed to severe ICI-associated myocarditis (9,800 pg/mL; IQR: 8,600–13,804 pg/mL) compared to those who achieved recovery (3,500 pg/mL; IQR: 2,310–5,200 pg/mL; $P = 0.003$). The comprehensive distribution patterns of these biomarkers are illustrated in Fig. 3.

Binary logistic regression analysis, incorporating NT-proBNP, cardiac troponin T (cTnT), creatine phosphokinase (CPK), creatine kinase MB isoenzyme (CK-MB) and age ($\leq 65$ years *vs.* >65 years), revealed that elevated NT-proBNP levels (OR 4.3, 95% CI

**Table 1 Baseline demographic and clinical characteristics of patients with severe *vs.* non-severe myocarditis.**

|  | Severe patients (*n* = 33) | Non-severe patients (*n* = 38) | *P*-value |
|---|---|---|---|
| Characteristics |  |  |  |
| Age, y | 70 (60, 75) | 69 (61, 74) | 0.205 |
| Male | 23 (69.7) | 27 (71.1) | 0.901 |
| Smoker | 14 (42.4) | 16 (42.1) | 0.978 |
| Hypertension | 10 (30.3%) | 6 (15.8%) | 0.144 |
| DM | 7 (21.2%) | 4 (10.5%) | 0.215 |
| CHD | 3 (9.1%) | 2 (5.3%) | 0.529 |
| Cancer Type |  |  |  |
| Lung cancer | 14 (19.7) | 15 (21.1) | 0.814 |
| Gastric cancer | 6 (8.5) | 4 (5.6) | 0.497 |
| Esophageal cancer | 4 (5.6) | 4 (5.6) | 1.000 |
| Other types | 9 (12.7) | 15 (21.1) | 0.322 |
| Line of Therapy |  |  |  |
| First line | 21 (29.6) | 20 (28.2) | 0.471 |
| Second line | 8 (11.3) | 11 (15.5) | 0.794 |
| Third line and beyond | 4 (5.6) | 7 (9.8) | 0.527 |
| Immunotherapy drugs |  |  |  |
| Sintilimab | 8 (11.3) | 6 (8.5) | 0.390 |
| Camrelizumab | 4 (5.6) | 10 (14.1) | 0.151 |
| Other drugs | 11 (15.5) | 22 (31.0) | 0.056 |
| Presence of other irAEs | 18 (25.4) | 14 (19.7) | 0.157 |
| Multiple other irAEs | 8 (11.3) | 2 (2.8) | 0.037 |

**Notes.**

Data are presented as medians (Q1, Q3) or n (%), as appropriate.

Abbreviations: DM, diabetes mellitus; CHD, coronary heart disease; BMI, body mass index; irAEs, immune-related adverse events.

[1.2–21.9]; $P = 0.023$) are an independent predictor of adverse outcomes in patients with severe myocarditis (Fig. 4).

## Treatment strategies and patient outcomes

The treatment regimens and clinical outcomes for patients with myocarditis of varying severities, including the use of immunosuppressive agents and adjunctive therapies, are summarized in Table 2. Among 38 patients diagnosed with non-severe myocarditis following discontinuation of ICI therapy, primary treatment involved oral prednisone (1 mg/kg/day) or intravenous methylprednisolone (1 mg/kg/day), with adjustments based on individual patient response and tolerance. Adjunctive medications included metoprolol, coenzyme Q10, and trimetazidine. During a two-month follow-up, 34 patients demonstrated symptom resolution and normalization of cardiac biomarkers; the remaining four exhibited persistently elevated cardiac cTnT levels, which normalized by six months.

In this study, 33 patients were diagnosed with severe myocarditis (Grades 3–4), and 18 of these patients died. The median time from diagnosis to death among the 18 deceased

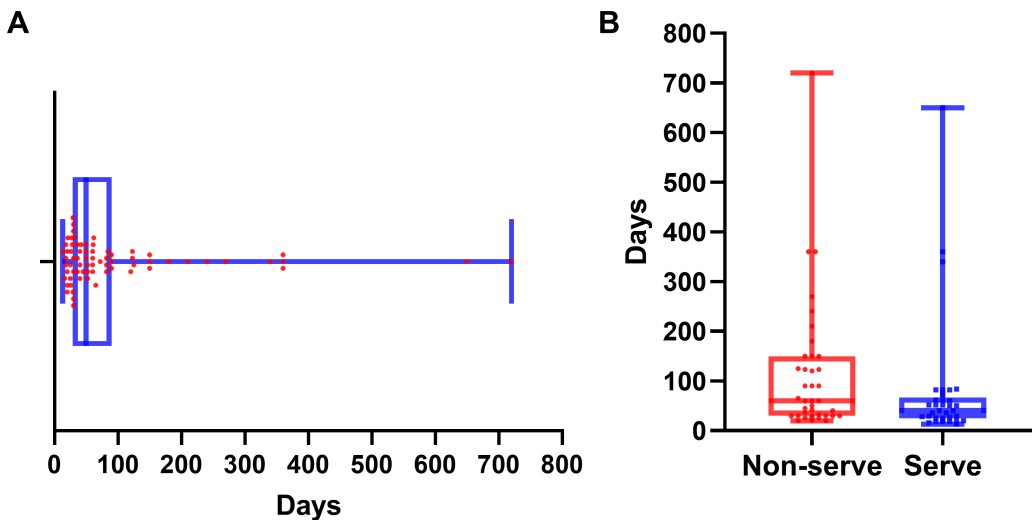

**Figure 2** **Time to onset of myocarditis following immune checkpoint inhibitor therapy.** (A) Distribution of time to myocarditis onset for all patients. (B) Comparison of time to myocarditis onset between the severe and non-severe myocarditis groups. The median is indicated by the central line, with the first and third quartiles represented by the box edges. Whiskers show the range, excluding outliers.

**Table 2** Treatment strategies for patients with ICI-associated myocarditis.

| Severity | Number of patients | Tumor type | Corticosteroid treatment | Combination therapy | Outcome |
|---|---|---|---|---|---|
| Non-severe | 38 | LC × 15, GC × 4, HCC × 4, TC × 3, EC × 3, Others (n = 6) | Oral prednisone or low-dose MP | – | All resolved |
| Severe | 18 | LC × 7, GC × 2, CCA × 2, TC × 2, EC × 2, Others (n = 3) | High-dose MP | – | 8 resolved, 10 deceased |
| Severe | 3 | LC × 3 | High-dose MP | IVIG | 1 resolved, 2 deceased |
| Severe | 2 | GC × 2 | High-dose MP | MMF | 1 resolved, 1 deceased |
| Severe | 2 | EC × 1, CCA × 1 | High-dose MP | Tocilizumab | All deceased |
| Severe | 8 | LC × 4, EC × 2, GC × 2 | High-dose MP | IVIG, Plasmapheresis, MMF | 5 resolved, 3 deceased |

**Notes.**

Abbreviations: MP, Methylprednisolone; IVIG, Intravenous Immunoglobulin; MMF, Mycophenolate Mofetil; LC, Lung cancer; GC, Gastric cancer; EC, Esophageal cancer; TC, Thymic Carcinoma; HCC, Liver cancer; CCA, Cholangiocarcinoma.

patients ranged from 2 days to 2 months. Among the 33 patients with severe myocarditis, 18 received high-dose intravenous methylprednisolone (500–1,000 mg per dose) as their sole immunosuppressive therapy, in conjunction with standard medications for managing arrhythmia or heart failure. Of these 18 patients, eight showed a favorable response, while the remaining 10 succumbed to the disease; seven deaths were directly due to myocarditis (six cases of Grade 4 and one case of Grade 3), and the other three fatalities were due to complications unrelated to myocarditis.

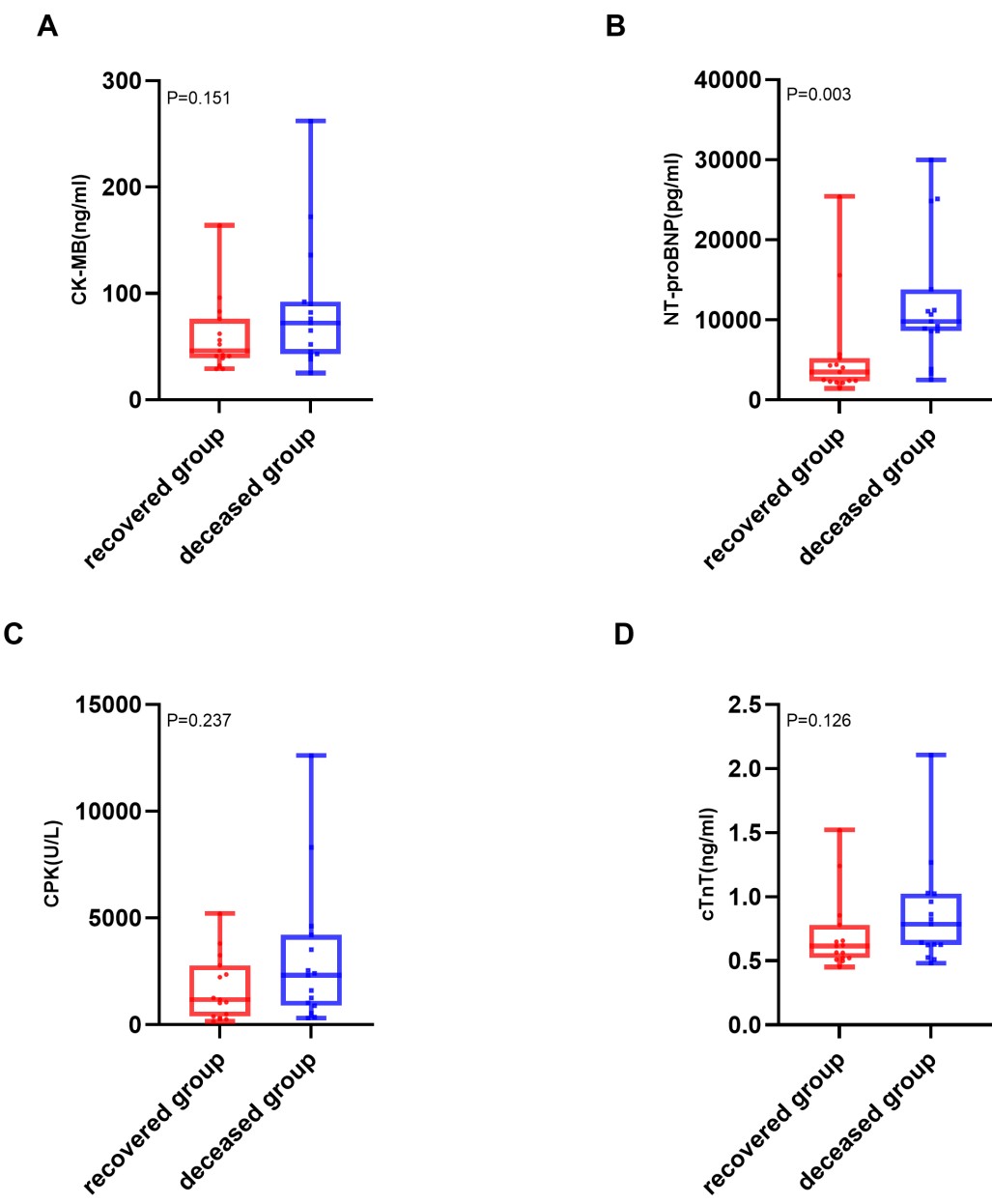

**Figure 3** **Comparison of cardiac biomarker levels between deceased and recovered patients with severe ICI-associated myocarditis.** (A) Creatine kinase-MB (CK-MB) levels (ng/mL) in the deceased and recovered groups. (B) N terminal pro-brain natriuretic peptide (NT-proBNP) levels (pg/mL) in the deceased and recovered groups. (C) Creatine phosphokinase (CPK) levels (U/L) in the deceased and recovered groups. (D) cardiac troponin T (cTnT) levels(ng/ml) in the deceased and recovered groups. The median is indicated by the central line, the box edges represent the first and third quartiles (IQR), and the whiskers extend to the most extreme data points that are not considered outliers.

Additionally, seven patients with severe myocarditis were treated with methylprednisolone in combination with other therapeutic modalities. Two patients with Grade 3 myocarditis achieved favorable outcomes after receiving intravenous methylprednisolone

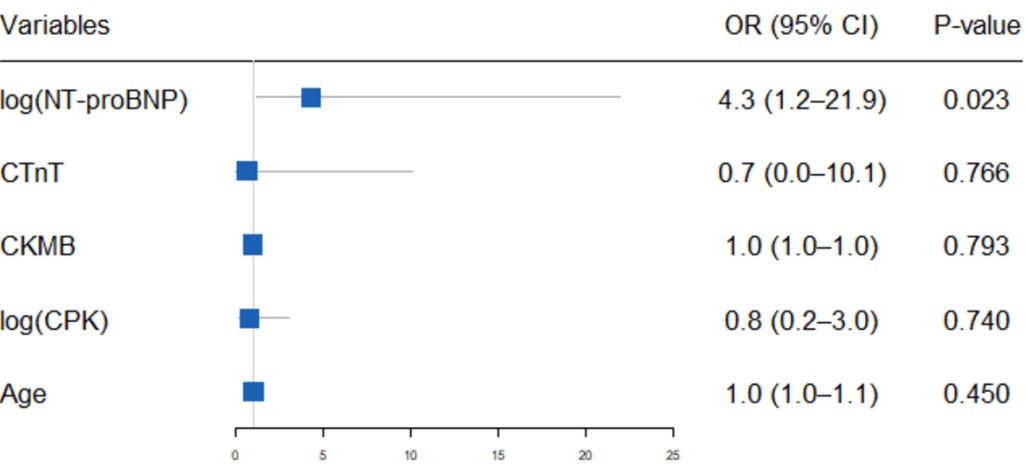

**Figure 4  Binary logistic regression of predictors for mortality in patients with severe ICI associated myocarditis.** The forest plot presents the odds ratios (ORs) and 95% confidence intervals (CIs) from a binary multivariable logistic regression analysis evaluating the association between various predictors and mortality in patients with severe immune checkpoint inhibitor (ICI)-associated myocarditis. The predictors include age, creatine kinase MB (CKMB), cardiac troponin T (cTnT), creatine phosphokinase (CPK), and N-terminal pro-B-type natriuretic peptide (NT-proBNP). Each predictor's OR is plotted on a logarithmic scale, with horizontal lines representing the 95% CIs.

combined with either intravenous immunoglobulin (IVIG; 2 g/kg) or mycophenolate mofetil (MMF; 1.5 g/day). However, five patients with Grade 4 myocarditis, despite receiving combination therapies, succumbed to their illness within a month.

Of particular note, eight patients with Grade 4 myocarditis were rapidly admitted to the cardiac intensive care unit (CICU) following MDT evaluation. In the CICU, they received comprehensive treatment that included intravenous methylprednisolone (1 g/day), IVIG at 2 g/kg, and MMF at 1.5 g/day. Additionally, plasmapheresis was specifically incorporated into the regimen for these critically ill patients. This procedure aimed to rapidly remove circulating autoantibodies and inflammatory mediators contributing to myocardial damage. Five of these patients recovered from myocarditis and were able to resume antitumor therapy.

## ICI resumption

In the current study, after a comprehensive evaluation by the MDT and thorough communication with the patients, nine patients were rechallenged with ICIs after their ICI-associated myocarditis was resolved, including six patients with Grade 1 and three patients with Grade 2 myocarditis. All patients were monitored closely and contacted the doctor if they felt unwell at any time. Seven patients did not experience irAEs, including cardiac toxicity, after rechallenge with ICIs. Two patient who developed Grade 2 myocarditis previously exhibited chest pain with significant elevation of troponin I when rechallenged with ICIs. The patient's symptoms and abnormal cardiac biomarkers were resolved after treatment with high-dose steroids and IVIG.

## DISCUSSION

ICIs have significantly enhanced clinical outcomes across various cancer types and are increasingly being employed in the early stages of disease and in combination with other therapies (*Walia & Prasad, 2023*). However, with the widespread use of ICIs, severe adverse events have been increasingly reported (*Sullivan & Weber, 2022*). Among these adverse events, ICI-associated myocarditis has attracted significant attention owing to its relatively low incidence and high mortality rate (*Munir et al., 2024*). In this study, patients diagnosed with ICI-associated myocarditis represented 1.02% of all patients who received ICI treatment at our center, which is notably lower than other adverse events, such as ICI-induced myositis (30.0%) and ICI-induced thyroiditis (25.0%).

In our cohort of patients diagnosed with ICI-associated myocarditis, the overall mortality rate directly attributable to ICI-induced cardiac toxicity was 21.1%. Notably, when stratified by disease severity, patients with Grade ≥3 myocarditis demonstrated a substantially elevated mortality rate of 54.5%, a finding that is consistent with previously reported outcomes in comparable patient populations (*Itzhaki Ben Zadok et al., 2023*; *Salem et al., 2018*).

Consistent with other studies, our research demonstrated that ICI-associated myocarditis typically manifests within three months of the initial dose (*Coustal et al., 2023*; *Xie et al., 2022*; *Qin et al., 2024*; *Thuny, Naidoo & Neilan, 2022*). Notably, severe myocarditis cases tend to emerge earlier in the course of ICI therapy than non-severe cases. This observation highlights the need for close clinical and laboratory surveillance, particularly during the first few weeks after ICI administration. The underlying mechanisms for this temporal pattern remain unclear, but may involve a more robust and rapid immune response in patients who develop severe myocarditis.

In contrast to the early emergence of severe myocarditis in some patients, the onset of ICI-associated myocarditis can be significantly delayed in some cases. In our study, we observed several cases in which patients were diagnosed with ICI-associated myocarditis more than 12 months after the commencement of therapy, with two notable cases occurring nearly two years after the initiation of treatment.

These two patients initially received a combination of ICI and chemotherapy as first-line therapy, followed by maintenance therapy with ICI. They presented with symptoms of chest tightness and elevated cTnT levels, which prompted an initial consultation with the cardiology service. Given the significant time interval since the initiation of ICI therapy, the possibility of ICI-associated myocarditis was initially overlooked, leading to the administration of metoprolol and trimetazidine in consideration of non-infectious myocarditis. Symptoms of chest tightness gradually worsened, and the cTnT level significantly increased, with new elevations in myoglobin and creatine kinase observed over the course of several days following treatment.

After a thorough review of the patients' clinical presentations, laboratory data, electrocardiographic findings, imaging examinations, and medical histories, we determined that ICI-associated myocarditis was the leading diagnostic consideration for both the patients. Following three days of treatment with 500 mg methylprednisolone daily,

both patients experienced marked improvement in clinical symptoms, accompanied by a significant reduction in serum cardiac biomarker levels. Eventually, the patient's symptoms disappeared and the abnormal cardiac biomarker levels normalized within a month. The potential for late-onset myocarditis highlights the necessity of vigilant long-term monitoring and a thorough diagnostic evaluation, regardless of the timing from the initiation of ICI treatment.

Peripheral tolerance mechanisms, such as the PD-1–PD-L1 signaling axis, are essential for maintaining immunological homeostasis in the cardiovascular system (*Munir et al., 2024*). Disruption of this pathway, such as that caused by ICI treatment, can lead to heightened autoimmune responses, potentially leading to inflammatory cardiomyopathies including myocarditis. Given their potent anti-inflammatory and immunosuppressive effects, glucocorticoids are considered the cornerstone of therapy for irAEs associated with ICIs, including myocarditis (*Thompson et al., 2022*; *Palaskas, Siddiqui & Deswal, 2024*).

Notably, thymic carcinoma (TC) was observed in five patients (7.0%) of our cohort (Table 2), a proportion higher than that reported in other studies of ICI-associated myocarditis. This finding aligns with the hypothesis that thymic tumors may represent a potential risk factor for immune checkpoint inhibitor-induced myocarditis, as suggested by prior research (*Fenioux et al., 2023*). The thymus's role in immune regulation and its proximity to the heart could contribute to heightened immune-mediated cardiac toxicity in these patients. However, the small sample size of TC cases in our study limits definitive conclusions, and further investigation is warranted to validate this association.

In our study, all patients with non-severe myocarditis received a low dose of steroids either alone or in combination with relevant cardioprotective medications. These patients, except for four, achieved full recovery within a two-month period. The cardiac troponin T (cTnT) levels in the four remaining patients gradually decreased over six months and ultimately returned to normal. This gradual normalization of cTnT levels may be attributed to ongoing immune activation due to the presence of immune checkpoint inhibitors or could indicate the presence of chronic myocarditis (*Vasbinder et al., 2022*; *Nishikawa et al., 2022*).

In patients with severe myocarditis, early initiation of high-dose corticosteroids is often recommended to reduce the incidence of major adverse cardiac events (MACE) (*Nielsen et al., 2024*; *Thuny, Naidoo & Neilan, 2022*). However, the efficacy of high-dose corticosteroid therapy as monotherapy for immunosuppression varies, and only a subset of patients experiences significant benefits.

In our study, among 18 patients with severe myocarditis treated exclusively with high-dose methylprednisolone as an immunosuppressive agent, along with medications for arrhythmia or heart failure, 11 succumbed to their condition. This highlights the potential insufficiency of single-agent treatment or resistance to high-dose corticosteroid therapy in this patient cohort. Therefore, there is an urgent need to explore alternative or adjunctive treatments to improve outcomes in patients with severe myocarditis.

High-dose corticosteroids in combination with agents such as high-dose intravenous immunoglobulin (IVIG), mycophenolate mofetil (MMF), and IL-6 inhibitors have been explored in the literature. However, these treatment regimens have not consistently

demonstrated satisfactory outcomes in patients with severe immune checkpoint inhibitor (ICI)-associated myocarditis (*Tedeschi et al., 2002*; *Bhat et al., 2023*; *Vicino et al., 2024*; *Chatzantonis et al., 2020*; *Doms et al., 2020*; *Campochiaro et al., 2021*).

In our study, seven patients received combined therapy, yet only two survived. Recently, a study involving 40 patients showed that the combination of ruxolitinib and high-dose abatacept significantly reduced the fatality rates associated with severe ICI-associated myocarditis (*Salem et al., 2023*). Despite the promising therapeutic effects observed with this dual-drug approach, large-scale prospective studies are necessary to further validate these findings.

Beyond pharmacological interventions, non-pharmacological approaches are also being explored to address the challenges posed by severe myocarditis (*Jo et al., 2024*). Plasma exchange (PEX) is a therapeutic procedure that rapidly removes antigens, antibodies, immune complexes, and cytokines from the plasma using a blood cell separation device (*Bauer et al., 2022*). By suppressing excessive cellular and humoral immune responses, PEX can prevent damage caused by overactive immunity. Rapid reduction of circulating immune mediators through PEX may provide immediate therapeutic benefits in patients with severe myocarditis, potentially improving outcomes and reducing mortality (*Ke et al., 2023*). CICUs play a crucial role in managing critically ill cardiac patients by providing advanced monitoring, life support, and specialized interventions in a multidisciplinary care environment (*Kasaoka, 2017*).

In the present study, eight patients with Grade 4 myocarditis were admitted to the CICU, where they received comprehensive care including plasma exchange as a critical component, alongside high-dose methylprednisolone, IVIG, and MMF. Five out of these eight patients survived, highlighting the potential significance of plasma exchange in enhancing outcomes specifically in the treatment of severe myocarditis. Plasmapheresis, as part of a multimodal regimen, showed a trend toward improved survival in this cohort. However, its role in ICI-associated myocarditis remains to be confirmed in larger studies.

In addition to developing effective treatment strategies, identifying predictive characteristics for adverse outcomes in patients with severe myocarditis is of paramount importance. Among the various biomarkers, NT-proBNP has emerged as a critical tool for assessing cardiac function and prognosis. Extensive research has consistently demonstrated that elevated NT-proBNP levels are strongly associated with poor outcomes in patients with acute myocarditis, reflecting the severity of myocardial injury and the degree of cardiac dysfunction (*Zhao et al., 2022*; *Abrar et al., 2016*). To contextualize its prognostic specificity, we compared 12 ICI-treated patients with cardiac symptoms but non-myocarditis diagnoses (*e.g.*, volume overload) to fatal myocarditis cases. Median NT-proBNP was significantly lower in controls (945 pg/mL, IQR 630–1,350) than in fatal myocarditis (13,804 pg/mL, IQR 8,600–21,200; $P < 0.001$), supporting its role in distinguishing severe myocarditis from other cardiac complications.

Notably, patients who succumbed to myocarditis exhibited markedly higher NT-proBNP levels compared to those who recovered, reinforcing its role as a robust prognostic marker. Binary logistic regression analysis further confirmed that elevated NT-proBNP levels were the only independent predictor of adverse outcomes (OR = 4.3, 95% CI

[1.2–21.9], $P = 0.023$). Unlike cTnT and CK-MB, which primarily reflect acute myocyte necrosis, NT-proBNP serves as a well-established biomarker of ventricular wall stress and neurohormonal activation (*Hall, 2005*). Its elevation is observed in various forms of myocardial inflammation and dysfunction, including fulminant myocarditis and heart failure, and is driven by increased hemodynamic burden as a result of heightened ventricular wall stress and systemic neurohormonal activation (*Cao, Jia & Zhu, 2019*). The activation of the sympathetic nervous system and the renin-angiotensin-aldosterone system (RAAS), along with endothelial dysfunction and cytokine-mediated inflammation, further amplifies this process (*Manolis, Manolis & Manolis, 2023*; *Sunayama et al., 2023*). Recent studies have shown that endothelial activation and fibroblast remodeling play key roles in the progression of myocardial inflammation and fibrosis, contributing to the sustained elevation (*Xuan et al., 2022*).

Although NT-proBNP levels were not systematically followed in most patients, two cases demonstrated a marked elevation in NT-proBNP coinciding with acute clinical deterioration during steroid tapering, suggesting that dynamic changes in NT-proBNP may reflect disease activity and prognosis. While this statistically significant association highlights the pivotal role of NT-proBNP in risk stratification and prognostic assessment for severe ICI-associated myocarditis, the wide confidence interval reflects substantial imprecision in the effect size estimate. This limitation stems primarily from the relatively small number of fatal events ($n = 18$) within our cohort. These findings emphasize the critical need for validation in larger, prospective studies to precisely define the predictive value of NT-proBNP and establish robust risk stratification thresholds.

Many previous studies many studies have discussed rechallenge with ICIs after the resolution of irAEs (*Chennamadhavuni et al., 2022*; *Scheiner et al., 2023*; *Guo et al., 2022*; *Jin et al., 2024*). However, only a few case reports have documented rechallenge with ICIs in patients who developed myocarditis (*Peleg Hasson et al., 2021*; *Menachery et al., 2023*).

In the current study, seven of nine patients who were rechallenged with ICIs did not experience any recurrent immune-related adverse events (irAEs), suggesting that the risk of recurrence is relatively low. However, two patient who had previously experienced Grade 2 myocarditis developed a recurrence of symptoms, including chest pain and elevated troponin T levels, upon rechallenge. This case highlights the potential risks associated with the rechallenge and underscores the importance of a cautious approach. MDT discussions are crucial in making these decisions, as they ensure a comprehensive evaluation of the patient's condition and a thorough discussion of the risks and benefits.

Despite providing valuable insights into ICI-associated myocarditis, this study has several limitations that need to be considered when interpreting the results. First, this was a retrospective study, which may be subject to incomplete or inaccurate recorded data. Additionally, the retrospective design made it difficult to control for all potential confounding factors, potentially introducing bias. Specifically, we were unable to fully account for baseline comorbidities (*e.g.*, diabetes mellitus, hypertension, chronic kidney disease), cancer stage (early *vs* advanced), and concomitant use of cardiotoxic medications (*e.g.*, anthracyclines, tyrosine kinase inhibitors). While we adjusted for major demographic and clinical variables in our multivariate models, residual confounding cannot be ruled out.

Future prospective studies should aim to comprehensively capture these variables to better understand their influence on cardiac biomarker dynamics and clinical outcomes in patients with ICI-associated myocarditis. Second, this study was conducted at a single center with a limited sample size, which may lack representativeness. Therefore, the generalizability of the findings may be limited and requires validation in multi-center studies with larger samples. Third, the diagnostic approach presents limitations. The absence of histopathological confirmation *via* EMB, the current diagnostic gold standard, combined with the partial reliance on CMR as the primary diagnostic modality, impacts diagnostic certainty. Although CMR demonstrated high utility in 34 patients (47.8%), its unavailability in 37 patients due to contraindications or clinical urgency necessitated diagnosis based on clinical criteria, biomarkers, ECG, and echocardiography. This diagnostic heterogeneity between CMR-confirmed and clinically diagnosed cases could potentially affect both the consistency of diagnostic accuracy and the interpretation of treatment outcomes. Fourth, the subgroup analyses-such as those examining treatment modalities and ICI rechallenge-involve very small numbers of patients. For instance, only eight patients received plasmapheresis among those with Grade 3–4 myocarditis, and only nine patients were rechallenged with ICIs after recovery. The reduced statistical power in these subgroups increases the risk of overinterpretation. The apparent survival benefit of plasmapheresis in Grade 4 myocarditis (5/8 survivors) is likely confounded by indication, as patients receiving this intervention were more critically ill and selected for intensive care. Fifth, the lack of a standardized treatment protocol across patients limits the ability to draw definitive conclusions regarding the efficacy of specific immunosuppressive strategies. Treatment decisions were made at the discretion of the attending physicians, resulting in considerable heterogeneity in regimens and timing of interventions. While this reflects the current state of clinical practice in the absence of evidence-based guidelines, it also underscores the need for future prospective studies to establish standardized therapeutic approaches. Finally, this study primarily focused on short-term clinical outcomes due to the retrospective design and limited long-term follow-up data available. Therefore, conclusions regarding long-term prognosis, recurrence risk, or late-onset complications could not be drawn. Future prospective studies with extended follow-up are needed to comprehensively assess the durability of treatment responses and the potential for late cardiac sequelae.

## CONCLUSIONS

ICIs have significantly improved cancer outcomes but are associated with severe adverse events, notably ICI-associated myocarditis, characterized by low incidence and high mortality. Our study underscores the critical role of NT-proBNP in identifying high-risk patients; elevated levels were independently predictive of poor prognosis, highlighting its importance for early risk stratification. Rechallenge with ICIs after myocarditis resolution is a viable option but must be approached cautiously. MDT discussions are essential to evaluate risks and benefits, ensuring informed decisions and close monitoring to promptly detect any recurrence.

### Funding
The authors received no funding for this work.

### Competing Interests
The authors declare there are no competing interests.

### Author Contributions
- Cheng He conceived and designed the experiments, prepared figures and/or tables, and approved the final draft.
- Linjuan Xu performed the experiments, prepared figures and/or tables, and approved the final draft.
- Zhihong Zhang analyzed the data, authored or reviewed drafts of the article, and approved the final draft.
- Jiong Wang conceived and designed the experiments, authored or reviewed drafts of the article, and approved the final draft.

### Human Ethics
The following information was supplied relating to ethical approvals (*i.e.*, approving body and any reference numbers):

The study protocol was approved by the Clinical Research Ethics Committee of the First Affiliated Hospital of the University of Science and Technology (Protocol Number: 2024-RE-288).

### Data Availability
The raw measurements are available in the Supplementary Files.

### Supplemental Information
Supplemental information for this article can be found online at http://dx.doi.org/10.7717/peerj.20020#supplemental-information.

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
