# Peer review of "NT-proBNP serves as a prognostic marker for adverse outcomes in severe immune checkpoint inhibitor-associated myocarditis"

_PeerJ, doi:10.7717/peerj.20020_

## Round 0.1 · original submission · Major Revisions

**Language Note:** The review process has identified that the English language must be improved. PeerJ can provide language editing services - please contact us at [email protected] for pricing (be sure to provide your manuscript number and title). Alternatively, you should make your own arrangements to improve the language quality and provide details in your response letter. – PeerJ Staff

·

Basic reporting

Basic reporting is good, and I have no major concerns.

-One suggestion to improve the transparency of data visualization is to plot the individual data points on top of the box plots to allow the reader to evaluate the full data.

-Lines 199-200: The phrasing is unclear as to how many patients were in the severe group and how many patients died.

-It would also be helpful to include the sample sizes in the text at lines 176-177.

Experimental design

I have no major concerns about experimental design, however, a few minor points should be addressed:

-Follow-up time should be specified in the methods for the whole cohort and specifically for the re-challenge group.

-Please discuss whether any patients underwent a biopsy or cardiac MRI to confirm the diagnosis, as biopsy is currently the gold standard in the diagnosis of ICI-myocarditis.

Validity of the findings

My only major concern with the article is that the conclusions regarding plasma exchange are overstated. This is a retrospective study with a small number of patients, and all the patients who received plasma exchange were also cared for in the CICU. I don't know that the differences in survival as documented in Table 2 have any potential generalizable significance. This is overstated in both the abstract and discussion sections.

Additional comments

I would suggest two additional points:

1- There seems to be a high number of thymic carcinoma patients in your cohort. This is interesting as thymic tumors are potentially a risk factor for ICI-myocarditis (PMID: 37884625). I would consider adding thymic carcinoma to the comparisons of severe vs non-severe in Table 2 and adding a comment about the high number of patients with this tumor type.

2- Were the deaths seen in this cohort that were not attributed to ICI-myocarditis attributed to the patient's cancer? This possibly raises the difficult point of the risk-benefit of ICI in the setting of advanced, life-threatening cancer.

Overall, this is a nice retrospective study that adds to the growing understanding of ICI-myocarditis clinically.

Reviewer 2 ·

Basic reporting

- Abstract: Clarify whether the 54.5% mortality refers specifically to the 33 patients with Grade 3–4 myocarditis.

- Introduction and references are appropriate, though consider citing large registry data (2022–2024) on ICI‐associated myocarditis.

- Decrease the number of significant figures in many numbers reported (Ex. line 185: OR = 4.33, 95% CI [1.21, 21.94] is easier to read as OR 4.3 CI 1.2, 21.9.

Experimental design

- Specify how many underwent biopsy or CMR versus biomarker‐based and multidisciplinary team diagnosis to assess potential misclassification.

- Clarify the differing ORs for NT‐proBNP (Abstract: 5.82; main text: 4.33) by indicating which model (univariable vs. adjusted) produced each.

- The apparent benefit of plasmapheresis in Grade 4 patients (5/8 survived) is intriguing but likely confounded by indication - note this in limitations.

Validity of the findings

- NT‐proBNP’s association with mortality is supported, but wide confidence intervals (e.g., 1.21–21.94) reflect limited precision. Emphasize this in the Discussion.

- The survival advantage with plasmapheresis should be described as hypothesis‐generating rather than definitive.

Additional comments

Strengths: MDT confirmation of myocarditis; focus on a clinically important, high‐mortality complication; appropriate use of bias‐reduced regression.

Minor: Encourage depositing the de‐identified dataset in a public repository; improve consistency in “Grade 3” vs “grade 3” wording.

I would recommend encouraging the authors to deposit their raw data in a recognized repository.

Reviewer 3 ·

Basic reporting

-

Experimental design

-

Validity of the findings

-

Additional comments

o In the index report, the authors explore an important and emerging complication of immune checkpoint inhibitors (ICIs), immune-related myocarditis, a rare but life-threatening condition. With the increased use of ICIs, identifying prognostic markers is clinically essential. The analysis is timely, clinically relevant, and of great use to oncologists and cardiologists. Well-structured methodology, including ASCO guideline-based grading, appropriate biomarker comparisons, and regression analysis with adjustment for confounders, adds to the paper's worth and merit. The paper is well written and largely free of grammatical errors. I have minor concerns which need to be addressed:-

1) Given the retrospective study, single-centre design, one should acknowledge the inherent biases (e.g., selection, information) and inability to establish temporal causality. Naturally, this limits the generalizability, which needs to be established in diverse settings.

2) Power analysis has not been performed, and the Subgroup analyses, like treatment modalities and ICI rechallenge, involve very small numbers. Naturally, the reduced statistical power may lead to overinterpretation.

3) Another drawback is the lack of a standardized treatment protocol. The authors present wide heterogeneity in treatment regimens across patients, which complicates the interpretation of treatment efficacy.

4) Lack of follow-up with NT-proBNP values is another limitation. This misses trends or peak levels, which may better correlate with the outcome

5) The study does not include cardiac MRI or endomyocardial biopsy for definitive diagnosis or severity quantification, which are considered the gold standard.

6) The manuscript could more deeply explore why NT-proBNP, among all biomarkers, was predictive, especially in contrast to cTnT or CK-MB. In general, one would expect all biomarkers to go hand-in-hand and be raised together and not disproportionately. Expand discussion on ventricular strain and neurohormonal activation as underlying factors in elevated NT-proBNP. Elaborate on the role of endothelial dysfunction and inflammation as the driving factor behind the pathogenesis of myocarditis. Accordingly, you may elaborate on the various cytokine disarray surrounding endothelial dysfunction and inflammation in myocarditis. Consider including recent references on the topic. You may consider:
o Xuan Y, Chen C, Wen Z, Wang DW. The Roles of Cardiac Fibroblasts and Endothelial Cells in Myocarditis. Front Cardiovasc Med. 2022 Apr 7;9:882027. doi: 10.3389/fcvm.2022.882027. PMID: 35463742; PMCID: PMC9022788.

7) The study did not include a control group of ICI-treated patients without myocarditis. Comparing the outcomes and biomarker profiles of these two groups could have provided additional insights and strengthened the analysis.

8) The study attempted to control for various patient characteristics, but there may be other confounding factors, such as underlying comorbidities, cancer stage, or concurrent medications, that were not fully accounted for in the analysis.

9) The study primarily focused on short-term outcomes, with limited information on the long-term prognosis and potential for recurrence of ICI-associated myocarditis. Longer-term follow-up studies would be valuable to assess the durability of the observed treatment effects and the risk of late-onset complications.

Overall, the manuscript makes a valuable contribution by identifying NT-proBNP as a clinically relevant, accessible biomarker for mortality prediction in ICI-associated myocarditis. Some alterations will help improve its quality further.

---

## Round 0.2 · accepted · Accept

The concerns raised in the previous round of review have been reasonably addressed.

·

Basic reporting

The authors have addressed my concerns.

Experimental design

-

Validity of the findings

-

Reviewer 3 ·

Basic reporting

The authors have satisfactorily responded to most concerns raised during the prior review. I have no further comments

Experimental design

-

Validity of the findings

-